# Callous and Unemotional Traits as Precursors to the Development of Female Psychopathy

**DOI:** 10.3390/ijerph20186786

**Published:** 2023-09-20

**Authors:** Ana Raquel Cardoso, Maria João Costa, Ana Isabel Sani, Diana Moreira

**Affiliations:** 1Faculty of Human and Social Sciences, University Fernando Pessoa (UFP), Praça 9 de Abril, 349, 4249-004 Porto, Portugal; 38594@ufp.edu.pt (A.R.C.); 38113@ufp.edu.pt (M.J.C.); anasani@ufp.edu.pt (A.I.S.); 2Observatory Permanent Violence and Crime (OPVC), FP-I3ID, Fernando Pessoa University, 4249-004 Porto, Portugal; 3Research Center on Child Studies (CIEC), University of Minho, 4710-057 Braga, Portugal; 4Laboratory of Neuropsychophysiology, Faculty of Psychology and Educational Sciences, University of Porto, 4200-135 Porto, Portugal; 5Projecto Homem, Centro de Solidariedade de Braga, Rua do Alcaide 29/31 Cividade, 4700-024 Braga, Portugal; 6IPNP Health, Institute of Psychology and Neuropsychology of Porto, 4000-053 Porto, Portugal; 7Centre for Philosophical and Humanistic Studies, Centro Regional de Braga, Universidade Católica Portuguesa, Rua de Camões, 60, 4710-362 Braga, Portugal

**Keywords:** female psychopathy, callous and unemotional traits, delinquency

## Abstract

Objective: Conduct a systematic review concerning the literature that reflects whether the callous and unemotional traits present in childhood and/or adolescence are precursors in the development of female psychopathy in adulthood. Materials and Methods: A systematic review involved consulting three databases—EBSCO, the Web of Science, and PubMed—for peer-reviewed and quantitative studies within the period 2000–2023. Nine articles with quality of three and above were included. Results: The presence of callous and unemotional traits designates a group of youth that show characteristics associated with psychopathy, specifically when predicting a more severe and chronic pattern of antisocial behaviour. Children with high rates of callous and unemotional traits, who show symptoms of attention deficit hyperactivity disorder in combination with severe conduct problems, are most likely to show features associated with psychopathy. The multidimensional psychopathy construct is considered a better predictor of future and stable antisocial behaviour than the callous and unemotional traits alone model. Conclusions: According to the studies selected, the callous and unemotional traits in childhood seem to be precursors of female psychopathy in adulthood, but only because of the way they seem to enhance conduct problems, disruptive behaviour disorders, and, as a possible outcome, delinquency and antisocial traits, which may be precursors of future psychopathy.

## 1. Introduction

Psychopathy is a neuropsychiatric disorder marked by deficient emotional responses, lack of empathy, and poor behavioral controls, commonly resulting in persistent antisocial deviance and criminal behavior [1]. Additionally, psychopathy is an adult condition; therefore, children cannot be diagnosed with this psychopathology, but showing signs of callous and unemotional (CU) traits in youth can be a sign of an elevated risk of psychopathy in adulthood [2]. This specific topic is a matter involving a lot of controversy since identifying young children with certain personality traits is often considered problematic and risky due to its negative implications and stigma [3]. For this same reason, in most studies, like the present one, the term “psychopathic traits” is used as it pertains to specific traits and behaviours related to the dimensions of psychopathy [3].

There have been several ways in which psychopathy has been pictured. In 2009, the Triarchic Conceptualization of psychopathy was presented, and it was from this model that psychopathy was conceptualised as a set of dimensions, including disinhibition, boldness, and meanness [4]. Disinhibition reveals a propensity for impulsive and control problems; boldness indicates tolerance to danger and the unknown; and meanness reflects deficient empathy and disinterest in close attachments [4]. This last dimension is associated with CU traits since it displays emotional insensitivity towards others [4]. High CU traits are correlated with increased levels of proactive aggression and, consequently, comparable levels of reactive aggression such as high thrill-seeking tendencies [4].

The Triarchic Conceptualization of psychopathy reveals three dimensions, grandiose–manipulative, callous–unemotional (CU), and impulsive–irresponsible, which are traits that are associated with psychopathy; when evaluating high levels of CU traits in combination with impulsive–irresponsible and, to a lesser extent, grandiose–manipulative traits, it amounts to a significant childhood risk and poorer adult life outcomes [5]. Of these three traits, CU and impulsive–irresponsible are independent of each other, but possibly overlap in children, as in adulthood [6]. CU traits are associated with proactive aggression and, to a lesser extent, with reactive aggression, as well as the fact that empathy, specifically cognitive, has a strong role in reactive aggression [7]. Furthermore, CU traits are also significantly associated with being proactive and reactive and with total aggression [8].

Through the decades, there have been numerous validated measures and scales for psychopathy. One of the most used was created in 1991 by Hare, named Psychopathy Checklist–Revised (PCL-R), which evaluates not only Factor I and Factor 2 but also the interpersonal, affective, lifestyle, and antisocial facets derived from factor analyses [9]. There was a recent scale that measures psychopathic personality in children, named the Child Problematic Traits Inventory (CPTI), by Colins et al. [10]. This scale is relevant for the topic in question of this review, given that it assesses a constellation of traits that resemble the psychopathic personality in adolescence and adulthood in ages 3 to 12 [10]. This scale focuses on the following dimensions: interpersonal (e.g., grandiose–deceitful), affective (i.e., CU), and behavioural (i.e., impulsive–need for stimulation) [10].

When it comes to the sex differences associated with psychopathic traits in childhood, considering the stage of development in which the assessment is carried out, the traits associated with psychopathy occur earlier in boys than in girls [11]. In addition to the aspect of time when these traits appear, sex differences impact which traits are more prevalent. The differences found between males and females are limited to impulsivity, where boys tend to achieve higher scores [12]. Orue et al. [13] deny this, stating that girls score higher on the impulsive–irresponsible dimension, while boys score higher on CU traits. This is confirmed by the study of Hoyo-Bilbao et al. [14], which points out that boys score higher on the dimensions of CU and grandiose–manipulative traits, while girls obtain higher scores on impulsive–irresponsible traits. The grandiose–manipulative dimension predicted a dominant interpersonal behaviour in boys but not in girls, while, in girls, a dominant interpersonal style was predicted by the impulsive–irresponsible dimension [15]. It seems that in girls the impulsive–irresponsible dimension is predominant, as explained by Conradi et al. [16] that, from an attachment perspective, grandiose–manipulative and impulsive–irresponsible traits are positively related to attachment anxiety, while CU traits are not.

Through these statements, one can hypothesise the idea that callous and unemotional traits are not so prevalent in girls. Wright et al. [17] enhance this hypothesis, since in their study boys scored significantly higher than girls on CU traits but not on aggression at age 2.5; however, at age 5, boys scored significantly higher than girls on both CU traits and aggression [18].

CU traits are related, unlike delinquency, to delayed pubertal timing, in other words, the later age of menarche, meaning that menarche is reported as being delayed when girls have higher CU traits [19]. This may be something interesting to note because CU traits and the (later) transition to puberty may be particularly important for predicting girls’ adjustment in adolescence, meaning that it can be one way to understand girls’ initiation of delinquency within adolescence.

Since CU traits reflect deficits in empathy [20], the concepts of affective empathy and cognitive empathy are also necessary topics to consider in this subject. Cognitive empathy (or Theory of Mind) focuses on our ability to represent the intentions and purposes of others as well as their beliefs, while affective empathy refers to the ability to understand the emotions of others as well as respond to those same emotions and affective state appropriately [21]. Although they follow different developmental paths, it is assumed that, both psychologically and neurologically, these concepts interact with each other and undergo developmental changes together during childhood and adolescence [21]. There is a clear relationship between higher scores on psychopathic traits and lower cognitive empathy, and male adolescents with high and moderate levels of psychopathy only show low levels of affective empathy, while female adolescents exhibit lower affective empathy as well as cognitive empathy [22,23].

Models of the development of CU traits note that children with high levels of these traits have impaired reward and punishment processing, such as possessing a dominant reward-response style and deficient punishment processing [24]. CU traits are associated with the slower recognition of facial expressions, which means that high levels of these traits may correlate with deficits in processing speed when it comes to recognizing emotions [25].

When boys and girls are in a situation of early rupture related to the emotional deprivation of their parents, boys do not have available resources to protect themselves against psychopathy, while, in girls, the higher levels of empathy and perhaps the lower vulnerability to CU traits can be protective against psychopathy, especially when faced with the unfortunate situation of early separation from parents and the emotional deprivation of parents [26]. In addition to autism, the psychiatric condition most frequently associated with empathic dysfunction is psychopathy [27]. The ability to repeatedly seriously injure other individuals is considered an indicator of profound distress and an inappropriate response to another person’s suffering [28].

When talking about CU traits, we cannot avoid mentioning terms such as delinquency, since some studies suggest that children with high levels of psychopathic traits (like CU) demonstrate higher levels of conduct problems (CP) and, therefore, a high positive association with aggression [29,30]. Oppositional Defiant Disorder (ODD), Conduct Disorder (CD), and Deficit Hyperactivity Disorder (ADHD) are the three most frequent disruptive behaviour disorders (DBD) during childhood as well as adolescence [31]. This statement is confirmed by Blair et al. [32], who explain that ADHD, ODD, and = CU traits are expressed early. ADHD and CP are two separate conditions that can co-occur in children, meaning that children with CP may also have signs of ADHD.

Even though girls usually develop CP later, it can still be as severe as the CP identified in boys, with the same problematic long-term implications [33]. Furthermore, ODD appears to be a crucial precursor of adolescent CD in children with ADHD, since the risk of CD is three times higher in these children [34]. This is relevant to the topic, since there is a subgroup of youth with CP that has CU traits; however, when facing this relatively small group, there is a high risk of them developing antisocial personality disorder (ASPD) since adults with ASPD are way more likely to have expressed CP and CU traits in childhood [32]. In contrast, even though ADHD can coexist with CD, the association between these two is highly accounted for by the co-occurrence of ODD [35]. Ghosh and Sinha [31] explain, in their study, that most of their cases showed a usual pattern of ADHD symptoms developing into ODD and, later, progressing to CD with an overlay of traits at all stages.

To explore the trajectories of both CP and CU traits, it is also important to understand that levels of CP and CU tend to change over time during one’s youth, when they co-occur with the fearlessness temperament [36]. The fearlessness temperament usually englobes a tendency to show less physiological arousal (such as blood pressure and rate of respiration) to unknown people and situations and a negative emotion to negative stimuli and, therefore, is different from both CU traits and CP [36,37]. Klingzell et al. [36] explore the relation among CU traits, CP, and the fearlessness temperament in children and show that decreasing or increasing CP and CU traits equate to decreases and increases, sequentially, in fearlessness levels and psychopathic traits. Fearlessness in early childhood is connected with CU traits as well as CP, since children who display both CP and CU traits are more expected to have a fearless temperament than children with only CP [10,38,39,40]. Although temperament is normally stable over time, it can also change during development, which is why it may also be fundamental to research if levels of fearlessness differ over time between opposite developmental trajectories of CP and CU traits [36,41,42].

It is worth noting that CU traits do not significantly predict delinquent behaviour trajectories; specifically, it is necessary to separate the different dimensions of psychopathic traits and consider the interpersonal and behavioural traits of individuals [43]. This is also defended by Herrington et al. [44], who highlight that the presence of CU traits is not synonymous with CD, despite their strong association. Even though CU traits have a relevant impact on antisocial behaviour, it is thought that a multidimensional approach is more capable of predicting this type of behaviour, since the Ansel et al. [45] study says that CU traits are a multifaceted construct, with specific CU dimensions predicting differential aspects of antisocial behaviour; however, callousness only predicted aggression and no other forms of this type of behaviour.

Gender differences in concepts like biology and socialisation may contribute to differences in the psychological features of the construct of psychopathy as well as the behavioural manifestations in women, like emotional instability and attempts to control others [46]. Therefore, understanding and acknowledging the relationship between psychopathic traits and types of aggression (particularly relational aggression) in women is fundamental to improving the general understanding of antisocial behaviour and psychopathy in women [7]. On the one hand, relational aggression is the intent to damage another person’s social status/relationships through rumour spreading or ostracism [47]. Physical aggression, on the other hand, is often divided into two subgroups: proactive aggression, which refers to behaviours such as coercion and dominance without instant provocation, and reactive aggression, characterised as a less controlled display, with a lack of emotions and anger with a goal, like dominance and territoriality [48].

It is fundamental to mention parental styles, since Muratori et al. [49] mention that a lack of positive parenting styles as well as traumatic events during childhood, which have been linked to psychopathy in adulthood, may aggravate the levels of CU traits in children, confirming that there are several ways in which these traumas in children can develop into psychopathy in the future [50].

Despite being considered a risk factor for both sexes, boys are more likely to be born into large families—which may allow for or increase the probability of greater interaction with other aggressive individuals, providing the development of antisocial behaviour—while girls with traces of psychopathy tend to be part of more agitated and problematic families [51].

Cruel and severe parental discipline is considered a predictor of both factors involved in psychopathy: Factor 1, related to affective and interpersonal issues with predictors like having a fragile affective bond with one of the parents as well as indifference or neglect; Factor 2, which involves antisocial behavioural characteristics with predictors like inadequate supervision during youth [50]. Therefore, it is possible to confirm that family dysfunction is considered a more significant psychological risk factor in psychopathy [52]. Furthermore, both negative and positive parenting may be less influenceable than previously thought in explaining some nonadaptive behaviours associated with psychopathic traits (such as risky sexual behaviours, substance use, and CP) [53]. On the contrary, the study by Dotterer et al. [54] confirms that low levels of parental affection and the levels of severe parenting (of both the father and mother) are associated with traces of CU traits in adolescents.

There is, in fact, a relationship between childhood trauma, specifically abuse, and the development of symptoms of psychopathy [55]. While it is suggested that childhood abuse and bonding problems may be associated with later psychopathy, with this association being stronger among girls, emotional and sexual abuse are not correlated with psychopathic traits [56,57]. Lindberg et al. [58] study violent offending girls and conclude that they showed more unstable interpersonal relationships as well as impersonal sexual behaviour, that they had been significantly more often sexually abused as children, and that they had more often been subjects of physical violence in the childhood home than boys. Childhood trauma, such as sexual abuse, tends to be highlighted as distorting to female development [58].

Since CU traits have been, for a lot of decades, the focus for assessing and investigating psychopathic traits in children [59], it is relevant to study if CU traits are precursors of the development of female psychopathy by exploring the possible connection between CU traits and delinquency and, possibly, its impacts on the development of psychopathic traits in adulthood.

## 2. Materials and Methods

To answer the guiding question of this study, “Can the callous and unemotional traits, verified in childhood and adolescence, be considered as precursors of the development of female psychopathy in adulthood?”, the most relevant studies published in the English language from 2000 to 2023 were analysed to focus on more recent studies, using the EBSCO, PubMed, and Web of Science databases. The search expression for all databases was the following: TI callous–unemotional traits AND TI (psychopath* OR delinq*) AND TX (girls or women or females or young woman or girl or female or young women) NOT TX psychopathol*.

To obtain the desired articles with greater empirical evidence to answer the research question, the terms Scientific Journals (Peer-Analysed) and Academic Journals were used, as inclusion and exclusion criteria were selected. Regarding the inclusion criteria, the review of articles includes studies that are related to children and adolescents, the female gender, callous and unemotional traits, psychopathy and psychopathic traits, delinquent behaviour, and delinquency. Regarding the exclusion criteria, studies only related to the male gender, case studies, substance abuse, neurology and neuropsychology, white-collar crimes, and validation of assessment instruments were excluded.

The analysis of the studies followed two stages; initially, both the title and abstract of the articles were analysed, and, afterwards, reading the previously selected articles in full was prioritised; lastly, the inclusion and exclusion criteria already mentioned were obeyed. For a better portrayal of the data selection and analysis steps, the Preferred Reporting Items for Systematic Reviews (PRISMA) flowchart guideline [60] was utilised, as seen in Figure 1.

The selection of articles was performed by two independent judges, and, as differences and conflicts arose in this selection of material, a consensus was achieved.

The Quantitative Research Assessment Tool (QRAT; Child Care & Early Education Research Connections, 2019) was used to assess the methodological quality of the studies included in this review. The QRAT comprises 12 items pertaining to the methodological features of the studies. Items can be rated −1, 0, 1, or NA (not applicable), except for the 12th item, where NA is not an option. According to the QRAT specifications, studies with lower scores should be regarded with more caution, compared to studies with higher scores, which are methodologically more robust. Most studies included in this review (66.6%) had a score of three or above.

## 3. Results

Previously, 36 studies were present; however, after removing duplicates (*N* = 13), 23 studies remained. These 23 studies were divided by the authors to analyse in full and then the decision was made to include 11 studies that abided by our inclusion and exclusion criteria (12 articles were excluded).

Regarding the total number of articles searched for this review (*N* = 11), only 81.8% of these studies were included according to the inclusion and exclusion criteria. After this, the final number of articles included was nine. Table 1 presents the publications selected according to the chosen inclusion criteria and to the reading of the articles in full, highlighting their titles, authors, years of publication, objectives, and results.

## 4. Discussion

Using the nine articles included in this review, it was possible to analyse them and then extract their main ideas, to verify and compare their discussions. The present systematic review aimed to understand if callous–unemotional traits are precursors of the development of female psychopathy by understanding whether CU traits are correlated with delinquency and if they, possibly, positively impact the development of psychopathic traits in adulthood. To do so, the present discussion of the studies selected highlights the relation among CU traits, delinquency, and psychopathic traits.

### 4.1. Psychopathic Traits, Delinquency, and Antisocial Traits

Psychopathy, as explained, is a constellation of co-occurring traits. CU traits were chosen for investigation into if they are precursors to the development of psychopathy in adulthood because this psychopathic trait is frequently associated with delinquency [44], conduct problems (CP) [29], aggression [30], and lack of emotional sensitivity towards others [4]. Having these types of behaviours in childhood is, as predicted, a significant predictor for antisocial behaviour [65].

CU traits in childhood are, it seems, precursors to the development of psychopathy in adulthood. However, this might only be true because these traits seem to potentiate CP, which, in turn, potentiates Disruptive Behaviour Disorders (DBD) diagnosis, which includes Attention Deficit Hyperactivity Disorder (ADHD), Oppositional Defiant Disorder (ODD), and Conduct Disorder (CD) as well as long-lasting psychosocial problems such as psychopathic traits (grandiose–deceitfulness, impulsivity, and need for stimulation) [32,36,61]. As we know, it is the existence of CU traits that indicates a group of youth that show characteristics connected with psychopathy, specifically when predicting antisocial behaviour [62]. Children with CU traits and CP are at risk for having higher levels of aggression and delinquency, and, in girls, CU traits alone are a more significant predictor for delinquency [63].

Furthermore, children with CU traits also have a higher risk of developing Antisocial Personality Disorder (ASPD) [32,62]. This also happens with children with ADHD in co-occurrence with CP, who express traits associated with psychopathy [61]. Finally, youth with delinquent behaviour, DBD, and CP may be at a strong risk of exhibiting psychopathy in the future, since high levels of psychopathic traits demonstrate higher levels of CP [29,30]. Additionally, stability and change in psychopathic traits are associated with stability and change in CP (as well as CU traits) [36].

### 4.2. Gender Differences

Regarding psychopathic traits, girls tend to have higher levels of impulsive–irresponsible traits than boys [13,14,15], and, when they co-occur with CU traits, both reactive aggression and proactive aggression are predominant. An interesting finding is that in children with low levels of CU traits and high levels of impulsive–irresponsible traits, the type of aggression adopted is only one, reactive aggression [37], which means that it is the presence of CU traits that intensifies the occurrence of proactive aggression. Since this last type of aggression is premeditated, goal-oriented, and associated with a lack of empathy and remorse, it conforms to the designation of CU traits. Still, it is curious that CU traits have this much of an impact, since reactive aggression is an impulsive and emotional response to a perceived threat or provocation, and it is more conforming to the most frequent psychopathic traits in girls, impulsive–irresponsible traits.

In terms of the prevalence of CU traits, it is noticed that these traits seem to be significantly higher in boys than in girls [17]. This might happen because, typically, girls have higher levels of empathy and lower vulnerability to CU traits. These traits, being the affective–interpersonal dimension of psychopathy, reflect empathy impairments and, in girls with high CU traits, lower affective and cognitive empathy [20,26].

The combination of CP and CU traits and the combination of CP and psychopathic personality were both equally strong predictors among girls for stable CP, while the three-dimensional psychopathic construct was a stronger predictor of continuing childhood CP among boys [3]. Children with CU traits seem to exhibit high and stable levels of CP, ADHD symptoms, impulsivity, and narcissism [67], while, in girls, CU traits have a significant impact on the stability of CP [3].

### 4.3. Genetic and Environmental Factors

The presence of a large cavum septum pellucidum was associated, in an adult sample, with CP, proactive aggression, and psychopathic traits, such as antisocial behaviour [66]. The cavum septum pellucidum is a cavity between the two leaflets of the septum pellucidum, and an enlarged one, in youth, was considered a marker of non-normative brain development, yet the specific causes of a large cavum septum pellucidum remain unclear [66]. Even though youth with an enlarged cavum septum pellucidum were more likely to demonstrate DBD, specifically CD, and ODD, it was not the group with a large cavum septum pellucidum that had a severe form of DBD, meaning that youth with this anomaly did not show increased levels of aggression and psychopathic traits relative to other youth with disruptive behaviour disorders [66]. With CU traits and psychopathic traits, it was found that they did not contribute to significant differences between DBD youth with or without a large cavum septum pellucidum [66].

Due to this finding, it can be relevant to propose the idea that there may be certain biological factors that have an association with psychopathy traits, CP, and/or DBD, since we know that, in girls, factors like a later age of menarche are connected to the presence of higher CU traits [19].

On the other hand, it is believed that environmental factors such as childhood trauma play a significant role in the development of symptoms of psychopathy [55] and that youth with a history of witnessing violence exhibit restricted affect and a lack of remorse and empathy and engage in more severe and varied forms of antisocial behaviour [68]. While boys tend to be born into large families, which might increase the risk of increasing antisocial behaviour due to greater interaction with other aggressive individuals, girls with traces of psychopathy tend to be part of more agitated and problematic families [51].

Despite CU traits not being so prevalent in girls, what was gathered was that girls with high levels of these traits might have had a history of witnessing violence and childhood trauma, where there was an opportunity for these traits to strengthen, which led to constant engagement in antisocial behaviour without any concern for how that might impact others, which increased the risk for behaviour problems later in life [43,69,70].

In summary, while some authors find that genetic factors have a connection with psychopathy traits, specifically CU traits, others argue that these factors play a bigger role in the evolution of these traits in boys, while, in girls, environmental factors may be a more relevant predictor [71,72]. The highest heritability of CU traits was observed in boys, while, in girls, shared environmental influences seem to contribute to the etiology of these traits [73].

### 4.4. Evaluating CU Traits in Childhood

It was advised by different researchers that when studying psychopathy and psychopathic personality traits, a multidimensional psychopathic-personality-based approach is a more fitting approach than the CU-based approach alone [65]. According to López-Romero et al. [65], whose study focused on three-year-old to five-year-old children, the multidimensional approach can present more information about the impact of each trait and its co-occurrences on predicting the development of psychopathy in adulthood. Salekin and Andrershed [74] support this notion when explaining that the broader concept of psychopathy could help predict negative outcomes and that the psychopathic personality construct should be used in all its components as well as CD.

Previous studies only address that CU traits might be influenced by other psychopathic traits since they typically co-occur [75]. Colins et al.’s [75] study with five-year-old children revealed that, while conducting the Child Problematic Traits Inventory (CPTI), the relation between CU traits and aggressive behaviour might be due to co-occurrence with other psychopathic traits and that interpersonal traits, like grandiose–manipulative, are uniquely related to bullying, reactive aggression, and delinquency, sometimes even stronger than CU traits. Furthermore, it was also found, in a study with a sample of 12-year-old to 18-year-old participants, that the callousness dimension of CU traits has a relevant influence on delinquency since it moderates the relationship between happy victimisation (the inappropriate positive emotions following a transgression) and delinquency [64].

In conclusion, the analysed studies reflected the importance of looking not only at the development of CU traits but also the CP and ADHD diagnoses [32], meaning that there is a need for subtyping children with CP and CD. Even so, it is relevant to emphasise that CU traits are not synonymous with CP, antisocial behaviour, or delinquency [43,44], despite having a significant impact on them. Also, it is beneficial to have a multidimensional approach, since CU traits alone seem to not be enough to predict future psychopathy [32,65,73,76], and it is favourable to also look at the CP and ADHD diagnoses [32,73].

## 5. Conclusions

Due to its specific topic, the present study was intended to answer its central question by relating the development of delinquency and CU traits in youth with the increased possibility of developing psychopathy in adulthood in girls. It was found that CU traits during childhood are only considered precursors of female psychopathy in adulthood because of the way they enhance types of behaviours (like CP and DBD) that exhibit delinquency and antisocial traits, which, in turn, may be precursors of future psychopathy.

Regarding the weaknesses of the results obtained through the systematic review, it was found that most studies about CU traits as well as psychopathy and psychopathic personality, in general, rarely only focus on the female sex, even though they are included in the sample [77]. According to Tully et al. [78], even when the research focuses on female psychopathy, it was not only underpowered but also outdated. Most studies did not consider the differences between both sexes, when both were included in the sample, in their discussion, though that information could have been relevant for the present study. Furthermore, the agreement concerning the clinical utility of the Hare psychopathy measures to assess women’s risk of future offending and violence is insufficient [79]. The main notion is that gender differences in the prevalence rates and outcomes of psychopathic traits impact the assessment of psychopathy. Since most research regards males, applying this same research when assessing females may not be recommended [80]. In addition to this, it is also important to take into consideration the methodology of the studies included in this review, since most of these did not obtain longitudinal data and were missing follow-ups. These data would be fundamental to investigate the evolution of CU traits from childhood to adulthood.

Despite the limitations that were pointed out, now more than ever, there is a focus on psychopathy in general and also, gradually, a focus on CU traits in younger ages and women. This study touched on some topics like parental behaviour, gender differences, childhood trauma, empathy, and, most importantly, psychopathy, CU traits, and delinquent behaviour. We also studied and analysed each article in depth, which led us to a better understanding of how each of these variables affects the development of CU traits. To conclude, further investigation would be fundamental and relevant to better understand this field of research and confirm whether CU traits in childhood are precursors to the development of female psychopathy in adulthood.

## Figures and Tables

**Figure 1 ijerph-20-06786-f001:**
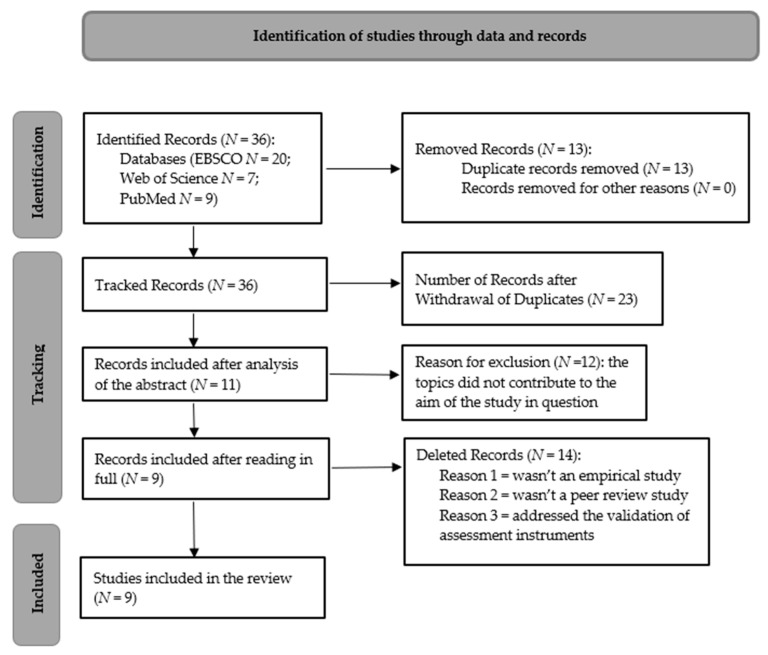
Flowchart of literature review process.

**Table 1 ijerph-20-06786-t001:** Summary of studies’ characteristics and the main findings.

Study ID	Objectives	Sample (Type, Age, % Girls)	Instruments	Main Findings
Barry et al. [61]	Use CU traits to identify a subgroup of children with ADHD and CP diagnosis (oppositional defiant disorder—ODD—or conduct disorder—CD—who show characteristics similar to adults with psychopathy.	*N* = 154 *M* = 8.36 (*SD* = 1.81) Girls = 22%	Diagnostic Interview Schedule for Children, Version 2.3 (DISC 2.3); Psychopathy Screening Device (PSD).	↑ CU traits showed features typically associated with psychopathy, such as a lack of fearfulness and a reward-dominant response style. Children w/ CU traits seemed less distressed by their CP. Impulsivity and antisocial behaviour alone are insufficient to document persons who fit the construct of psychopathy.
Frick et al. [62]	Test whether the presence of CU traits designates a group of children with CP who show an especially severe and chronic pattern of CP and delinquency.	*N* = 98 *M* = 12.36 (*SD* = 1.73) Girls = 47%	Antisocial Process Screening Device (APSD); Children’s Symptom Inventory-4 (CSI-4); NIMH Diagnostic Interview Schedule for Children-Version 4 (DISC-IV); Self-Report of Delinquency Scale (SRD).	Children with CU traits and CP showed ↥ rates of CP, self-reported delinquency, and parent-reported police contacts. Children with CP and CU traits showed an ↥ and stable rate of delinquency. It is the presence of CU traits, and not impulsivity alone, that designates a group of youth that show characteristics associated with psychopathy.
Frick et al. [63]	The role of CU traits and CP in predicting CP severity, severity and type of aggression, and self-reported delinquency.	*N* = 98 *M* = 12.36 (*SD* = 1.73) Girls = 47%	NIMH Diagnostic Interview Schedule for Children—Version 4 (DISC-4); Aggressive Behavior Rating Scale; Self-Reported Delinquency Scale—Youth self-report (SRD).	Children with CU traits + CP = ↑ number and variety of CP at follow-up than children with high levels of CP alone. They were also at risk for showing ↑ levels of aggression, especially proactive aggression, and self-reported delinquency. CU traits predicted self-reported delinquency in some. Girls with ↑ CU traits and without significant CP = ↑ predictive value of self-reported delinquency.
Frogner et al. [3]	Examine whether psychopathic traits and concurrent conduct problems (CP) predict future CP.	*N* = 1867 *M* = 3.86 (*SD* = 0.86) 1-year follow-up: *M* = 2.041 2-year follow-up: *M* = 1.934 Girls = 47%	Questionnaires completed by preschool teachers concerning their children during the last six months.	Psychopathic traits alone (without co-occurring CP) ⇎ continuing childhood CP. Psychopathic traits (GD, CU and INS) in combination with concurrent CP in both boys and girls = continuing childhood CP. CP + CU traits and CP + Psychopathic Personality = equally strong predictors for future CP among girls.
Klingzell et al. [36]	Investigate the joint development of CP and CU traits in early childhood.	*N* = 2031 Time 1: *M* = 3.86 (*SD* = 0.86) Girls = 46.9%	Ten items based on criteria of ODD and CD of the DSM-5; Child Problematic Traits Inventory (CPTI).	Stable ↑ CP and CU traits = ↑ levels of fearlessness, psychopathic traits, including grandiose–deceitfulness, and impulsivity, and need for stimulation. Children w/decreasing or increasing CP and CU traits = decreases and increases respectively in their levels of fearlessness and psychopathic traits. Stability and change in fearlessness and psychopathic traits = stability and change in CP and CU traits.
Kunimatsu et al. [64]	Examining associations among CU traits, happy victimisation, and delinquency.	*N* = 59 *M* = 14.98 (*SD* = 1.30) Girls = 100%	Self-Report of Delinquency Scale (SDR); Outcome Expectations Questionnaire (OEQ); Inventory of Callous-Unemotional Traits (ICU).	Positive associations among happy victimisation, CU traits, and delinquency. Happy victimisation and CU traits play significant roles in delinquent and violent behaviour.
López-Romero et al. [65]	Test predictive and incremental value of psychopathic trait dimensions in early childhood to predict future and stable CP and aggression in early childhood.	*N* = 2247 *M* = 4.25 (*SD* = 0.91) Girls = 48.6%	Child Problematic Traits Inventory (CPTI); EAS Temperament Survey for Children; Fast Track Social Competence Scale-Parent Version; Achenbach System of Empirically Based Assessment, Preschool Form (ASEBA); Parents and Teachers Report of Reactive and Proactive Behaviors.	Psychopathic personality + CP group = highest levels of grandiose–deceitful (GD) traits and CP. Psychopathic personality > CU traits in identifying children with CP at increased risk for future and stable CP, as well as future aggression. CU traits alone are less likely to display high CP at baseline, as well as future and stable negative outcomes.
White et al. [66]	Investigate the relationship between a large cavum septum pellucidum (CSP) and symptom severity in disruptive behaviour disorders (DBD).	*N* = 59 (32 youths with DBD and 27 healthy comparison youths) Youth with DBD: *M* = 14.90 (*SD* = 2.08) Healthy controls: *M* = 14.38 (*SD* = 2.44) Girls = 63%	Inventory of Callous-Unemotional Traits (ICU); Antisocial Process Screening Device (APSD); reactive/proactive rating scale.	Youth with a large CSP were more likely to meet criteria for DBD than youth without a large CSP. Individuals with a large CSP have an ↥ risk for aggressive behaviour, psychopathic traits, and having Oppositional Defiant Disorder (ODD) and CD. Large CSP ⇎ severe and ↥ risk for DBD. No significant differences in psychopathic/CU traits between DBD youth with and without a large CSP.

*Note*: APSD = Antisocial Process Screening Device; ASEBA = Achenbach System of Empirically Based Assessment; CD = Conduct Disorder; CP = Conduct Problems; CPTI = Child Problematic Traits Inventory; CSI-4 = Children’s Symptom Inventory-4; CSP = Cavum Septum Pellucidum; CU = callous–unemotional; DBD = Disruptive Behaviour Disorders; DISC 2.3 = Diagnostic Interview Schedule for Children—Version 2.3; DISC-IV = Diagnostic Interview Schedule for Children—Version 4; GM = grandiose–manipulative; II = impulsive–irresponsible; GD = grandiose–deceitful; ICU = Inventory of Callous-Unemotional Traits; INS = Impulsive/Need for Stimulation; ODD = Oppositional Defiant Disorder; OEQ = Outcome Expectations Questionnaire; PCL = Psychopathy Checklist; PCL-R = Psychopathy Checklist-Revised; PSD = Psychopathy Screening Device; SDR = Self-Report of Delinquency Scale; SOFIA = Social and Physical Development, Interventions and Adaptation.

## Data Availability

No new data was analyzed, collected or created in this study.

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
