# Peer review of "Callous and Unemotional Traits as Precursors to the Development of Female Psychopathy"

_ijerph, 2023, doi:10.3390/ijerph20186786_

Round 1

Reviewer 1 Report

Thank you for the opportunity to revise the manuscript “Callous and Unemotional Traits as Precursors to the Development of Female Psychopathy”. The proposal of the paper is to conduct a systematic review concerning literature that reflects whether the Callous and Unemotional (CU) Traits present in childhood and/or adolescence are, or not, precursors in the development of female psychopathy in adulthood.

Research on psychopathy increasingly focuses on the occurrence and the development of early onsets in childhood and adolescence. The term of callous-unemotional traits (CU traits) refers to characteristics, which are similar to the affective features of adult psychopathy. Extensive evidence suggests that high CU traits in childhood and adolescence can reliably identify individuals at risk for severe antisocial outcomes. For this reason, it is of great interest to detect symptoms as early as possible and my sense is that the paper could be of potential interest to the IJERPH readers.

The authors have provided a well-documented systematic review, with summarized tables of the studies described, which adheres to PRISMA guidelines also using the Quantitative Research Assessment Tool in order to assess the methodological quality of the studies enclosed in the review.

Below I offer some major and minor concerns based on my reading of the paper. While the topic is important and the scientific reasoning and methods are sound I have some major concerns:

In my opinion the introduction has to be shorter. The purpose of any literature review is to summarize and synthesize the arguments and ideas of existing knowledge. Thus, a review gives the direction to be headed for its success establishing among the authors’ in-depth understanding and knowledge of their field subject.  While the vast majority of research on contextual factors and CU traits was focused in the introduction, I think lacking models of the development of CU traits as the responsivity to rewards or the emotional processing. Findings from literature, for example, indicate that preschool children high on CU traits were less accurate in recognizing facial expressions.

This topic is not original population, but considering that only few studies included findings on the associations between ICU scores and internalizing psychopathology, and anxiety-related problems on female population, the present review is interesting.

And, compared with other published material, focusing on females population appears as an important research goal.

Figure 1 and Tables are appropriate but I suggest to delete Tables 1 and 2  at pp. 7 - 8 of the current manuscript reporting data along the text.

Paragraph “Evaluating CU Traits in Childhood” at p. 6: because of the relevance of age for these findings, I suggest to specify this information when Authors cited studies with references n. 67, 79, 80 and 81.

The discussion and conclusions are consistent with the evidence presented addressing the main research question of the review but

The conclusion is way too long. Please consider moving part of the first and the second paragraph to the discussion section.  While Authors reportet that “……did not consider the differences between both sexes, when both were included in the sample……” (p. 14) limitations must be acknowledged also considering the methodology of included studies and the lack of longitudinal data.

The references section is appropriate.

Author Response

We greatly appreciate the reviewer’s comments and suggestions. The introduction was shortened, and as suggested, an excerpt about the responsiveness to rewards and emotional processing in children with high levels of CU traits was included. When shortening the introduction some references were discarded. The authors deleted Tables 1 and 2, leaving along the manuscript explanation of the reporting data. The Paragraph “Evaluating CU Traits in Childhood” on p. 6 was modified, specifying the age of the population included in the studies mentioned in that same paragraph. The conclusion was shortened, moving the majority of the (original) first and second paragraphs to the discussion section just as recommended. More evidence about limitations regarding sex differences was added as well as the acknowledgement about the lack of longitudinal data.

We hope to have addressed all comments and issues mentioned since there is a complete agreement with all recommendations.

Reviewer 2 Report

Once the reader delves into this paper there is a considerable volume of good research.  The aim of the research was to identify the unsociable behaviors of children as a precursor to adult female psychopathy.  The research is a review of published data from a number of respectful sources but does not define psychopathy to position the research.   (Psychopathy is a neuropsychiatric disorder marked by deficient emotional responses, lack of empathy, and poor behavioral controls, commonly resulting in persistent antisocial deviance and criminal behavior.   https://www.ncbi.nlm.nih.gov/pmc/articles/PMC4321752/#:~:text=Psychopathy%20is%20a%20neuropsychiatric%20disorder,antisocial%20deviance%20and%20criminal%20behavior).  While the authors differentiate between psychopathic traits and behaviors related to the dimensions of psychopathy, the overuse of acronyms throughout the paper lessens the readability of the paper.  Throughout the paper the use of two or three letter acronyms is utilized extensively, and usually correctly, but causes reading difficulties.  While the three letter acronyms are suitable, the readability of the paper would be improved if the two letter acronyms were replaced with the actual wording of the behavior dimension.  There is no requirement to use the dimension or behavior name in conjunction with the acronym – one or the other is sufficient.  The usual rule is to provide the acronym where it first appears in the script, however the overuse of acronyms can spoil the readability of the paper.  There is an issue with formatting of the references which are inconsistent and require editing to present a common format.

As for the research, the introduction, research design, literature selection method, and the presentation of this data is excellent including the tables and figures.  The abstract summarizes the research very well, but the acronyms require replacement with the actual wording of the trait, dimension or behavior.  The introduction presents the relevant literature with a considered narrative to align this with the aims of the paper.  The methods and materials record in detail the literature selection method and tabulated data while the results provide a good explanation of the authors epistemology and ontology relative to the collected research data.  The discussion provides an in-depth interpretation of the data resulting in a plausible explanation that is inclusive of all the data collected.  The paper is eloquently and successfully presented in the conclusions including limitations of the research.  Overall, this is good research which is partially negated by the presentation which can be improved.

The requirement to produce a research paper in English in contrast to the authors native language does present issues for the authors – especially when the requirement is academic English.  A suggestion is to update the presentation to repair the acronym issue, then use a software editing program such as Microsoft Editor or the free version of Grammarly (there are other programs) to check the spelling and parsing of the paper.  However, if the authors do use editing software there is need to ensure that the meaning in the original manuscript is not lost or otherwise changed.

The requirement to produce a research paper in English in contrast to the authors native language does present issues for the authors – especially when the requirement is academic English.  A suggestion is to update the presentation to repair the acronym issue, then use a software editing program such as Microsoft Editor or the free version of Grammarly (there are other programs) to check the spelling and parsing of the paper.  However, if the authors do use editing software there is need to ensure that the meaning in the original manuscript is not lost or otherwise changed.

Author Response

We are grateful for the positive appreciation of our work and its consideration for publication. At the beginning of the introduction, as recommended, there is now a brief definition of psychopathy. Regarding the grammar, the acronym issue was resolved. Most of the two-letter acronyms were replaced with the actual word, however no changes to the acronyms CU (Callous-Unemotional), CP (Conduct Problems) and CD (Conduct Disorder) as they were the most predominant throughout the review. A software editing program, specifically Grammarly, was used to check the paper’s spelling, making sure the meaning of the sentences was not lost or modified. The acronyms in the abstract are now replaced with the actual wording or the trait, dimension or behavior. We now believe that the references are consistent and presented with a common format.

We completely agree with the recommendations and made a serious effort to fully meet all of those provided.

Round 2

Reviewer 1 Report

Authors have indicated that they had added new parts, in particular an excerpt of 6 lines about the responsiveness to rewards and emotional processing in children with high levels of CU traits was included (e.g.: p. 3 of the current form of the manuscript).

They had shortened the introduction section.

Tables 1 and 2, as suggested, are deleted leaving along the manuscript explanation of the reporting data.

More evidence about limitations regarding sex differences was included as well as the acknowledgement about the lack of longitudinal data.

Finally, the conclusion was shortened.

Globally, I think that the quality of the manuscript was improved. This creates a circunstance where I can recommand the publication.